# DYNAMIC COATTENTION NETWORKS
# FOR QUESTION ANSWERING

**Caiming Xiong,**[*] **Victor Zhong,**[*] **Richard Socher**
Salesforce Research
Palo Alto, CA 94301, USA
{cxiong, vzhong, rsocher}@salesforce.com

## ABSTRACT

Several deep learning models have been proposed for question answering. However, due to their single-pass nature, they have no way to recover from local maxima corresponding to incorrect answers. To address this problem, we introduce the Dynamic Coattention Network (DCN) for question answering. The DCN first fuses co-dependent representations of the question and the document in order to focus on relevant parts of both. Then a dynamic pointing decoder iterates over potential answer spans. This iterative procedure enables the model to recover from initial local maxima corresponding to incorrect answers. On the Stanford question answering dataset, a single DCN model improves the previous state of the art from 71.0% F1 to 75.9%, while a DCN ensemble obtains 80.4% F1.

## 1 INTRODUCTION

Question answering (QA) is a crucial task in natural language processing that requires both natural language understanding and world knowledge. Previous QA datasets tend to be high in quality due to human annotation, but small in size (Berant et al., 2014; Richardson et al., 2013). Hence, they did not allow for training data-intensive, expressive models such as deep neural networks.

To address this problem, researchers have developed large-scale datasets through semi-automated techniques (Hermann et al., 2015; Hill et al., 2016). Compared to their smaller, hand-annotated counterparts, these QA datasets allow the training of more expressive models. However, it has been shown that they differ from more natural, human annotated datasets in the types of reasoning required to answer the questions (Chen et al., 2016).

Recently, Rajpurkar et al. (2016) released the Stanford Question Answering dataset (SQuAD), which is orders of magnitude larger than all previous hand-annotated datasets and has a variety of qualities that culminate in a natural QA task. SQuAD has the desirable quality that answers are spans in a reference document. This constrains answers to the space of all possible spans. However, Rajpurkar et al. (2016) show that the dataset retains a diverse set of answers and requires different forms of logical reasoning, including multi-sentence reasoning.

We introduce the Dynamic Coattention Network (DCN), illustrated in Fig. 1, an end-to-end neural network for question answering. The model consists of a coattentive encoder that captures the interactions between the question and the document, as well as a dynamic pointing decoder that alternates between estimating the start and end of the answer span. Our single model obtains an F1 of 75.9% compared to the best published result of 71.0% (Yu et al., 2016). In addition, our ensemble model obtains an F1 of 80.4% compared to the second best result of 78.1% on the official SQuAD leaderboard.[1]

---

[*]Equal contribution

[1]As of Nov. 3 2016. See https://rajpurkar.github.io/SQuAD-explorer/ for latest results.

## 2 DYNAMIC COATTENTION NETWORKS

Figure 1 illustrates an overview of the DCN. We first describe the encoders for the document and the question, followed by the coattention mechanism and the dynamic decoder which produces the answer span.

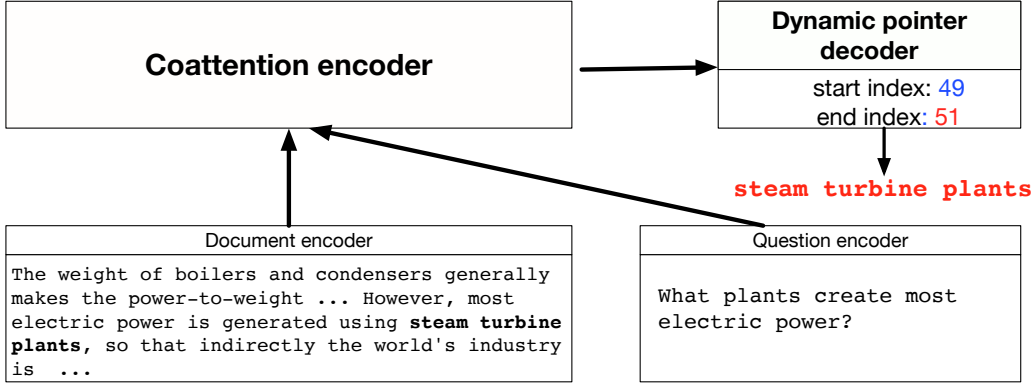

Figure 1: Overview of the Dynamic Coattention Network.

### 2.1 DOCUMENT AND QUESTION ENCODER

Let $(x_1^Q, x_2^Q, \ldots, x_n^Q)$ denote the sequence of word vectors corresponding to words in the question and $(x_1^D, x_2^D, \ldots, x_m^D)$ denote the same for words in the document. Using an LSTM (Hochreiter & Schmidhuber, 1997), we encode the document as: $d_t = \text{LSTM}_{enc}\left(d_{t-1}, x_t^D\right)$. We define the document encoding matrix as $D = [d_1 \ldots d_m \, d_\varnothing] \in \mathbb{R}^{\ell \times (m+1)}$. We also add a sentinel vector $d_\varnothing$ (Merity et al., 2016), which we later show allows the model to not attend to any particular word in the input.

The question embeddings are computed with the same LSTM to share representation power: $q_t = \text{LSTM}_{enc}\left(q_{t-1}, x_t^Q\right)$. We define an intermediate question representation $Q' = [q_1 \ldots q_n \, q_\varnothing] \in \mathbb{R}^{\ell \times (n+1)}$. To allow for variation between the question encoding space and the document encoding space, we introduce a non-linear projection layer on top of the question encoding. The final representation for the question becomes: $Q = \tanh\left(W^{(Q)} Q' + b^{(Q)}\right) \in \mathbb{R}^{\ell \times (n+1)}$.

### 2.2 COATTENTION ENCODER

We propose a coattention mechanism that attends to the question and document simultaneously, similar to (Lu et al., 2016), and finally fuses both attention contexts. Figure 2 provides an illustration of the coattention encoder.

We first compute the affinity matrix, which contains affinity scores corresponding to all pairs of document words and question words: $L = D^\top Q \in \mathbb{R}^{(m+1) \times (n+1)}$. The affinity matrix is normalized row-wise to produce the attention weights $A^Q$ across the document for each word in the question, and column-wise to produce the attention weights $A^D$ across the question for each word in the document:

$$A^Q = \text{softmax}\left(L\right) \in \mathbb{R}^{(m+1) \times (n+1)} \text{ and } A^D = \text{softmax}\left(L^\top\right) \in \mathbb{R}^{(n+1) \times (m+1)} \quad (1)$$

Next, we compute the summaries, or attention contexts, of the document in light of each word of the question.

$$C^Q = DA^Q \in \mathbb{R}^{\ell \times (n+1)}. \quad (2)$$

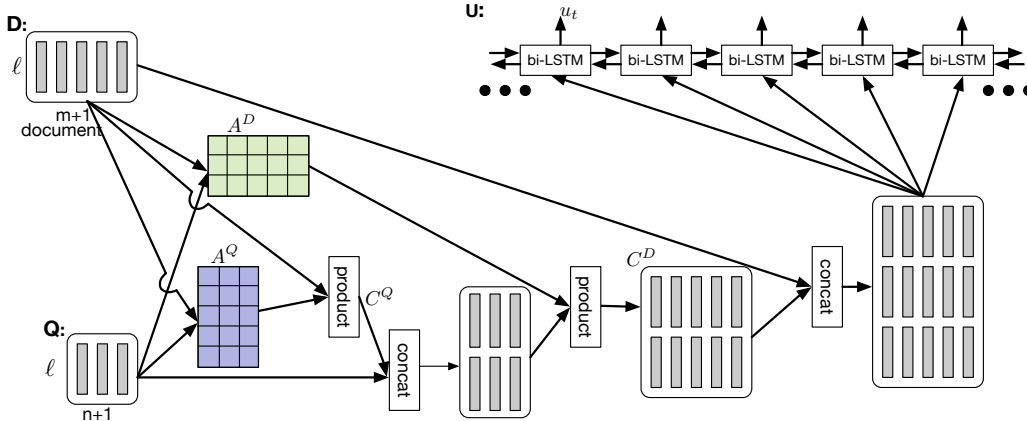

Figure 2: Coattention encoder. The affinity matrix $L$ is not shown here. We instead directly show the normalized attention weights $A^D$ and $A^Q$.

We similarly compute the summaries $QA^D$ of the question in light of each word of the document. Similar to Cui et al. (2016), we also compute the summaries $C^Q A^D$ of the previous attention contexts in light of each word of the document. These two operations can be done in parallel, as is shown in Eq. 3. One possible interpretation for the operation $C^Q A^D$ is the mapping of question encoding into space of document encodings.

$$ C^D = \left[ Q; C^Q \right] A^D \in \mathbb{R}^{2\ell \times (m+1)}. \tag{3} $$

We define $C^D$, a co-dependent representation of the question and document, as the coattention context. We use the notation $[a; b]$ for concatenating the vectors $a$ and $b$ horizontally.

The last step is the fusion of temporal information to the coattention context via a bidirectional LSTM:

$$ u_t = \text{Bi-LSTM}\left( u_{t-1}, u_{t+1}, \left[ d_t; c_t^D \right] \right) \in \mathbb{R}^{2\ell}. \tag{4} $$

We define $U = [u_1, \ldots, u_m] \in \mathbb{R}^{2\ell \times m}$, which provides a foundation for selecting which span may be the best possible answer, as the coattention encoding.

## 2.3 DYNAMIC POINTING DECODER

Due to the nature of SQuAD, an intuitive method for producing the answer span is by predicting the start and end points of the span (Wang & Jiang, 2016b). However, given a question-document pair, there may exist several intuitive answer spans within the document, each corresponding to a local maxima. We propose an iterative technique to select an answer span by alternating between predicting the start point and predicting the end point. This iterative procedure allows the model to recover from initial local maxima corresponding to incorrect answer spans.

Figure 3 provides an illustration of the Dynamic Decoder, which is similar to a state machine whose state is maintained by an LSTM-based sequential model. During each iteration, the decoder updates its state taking into account the coattention encoding corresponding to current estimates of the start and end positions, and produces, via a multilayer neural network, new estimates of the start and end positions.

Let $h_i$, $s_i$, and $e_i$ denote the hidden state of the LSTM, the estimate of the position, and the estimate of the end position during iteration $i$. The LSTM state update is then described by Eq. 5.

$$ h_i = \text{LSTM}_{dec}\left( h_{i-1}, \left[ u_{s_{i-1}}; u_{e_{i-1}} \right] \right) \tag{5} $$

where $u_{s_{i-1}}$ and $u_{e_{i-1}}$ are the representations corresponding to the previous estimate of the start and end positions in the coattention encoding $U$.

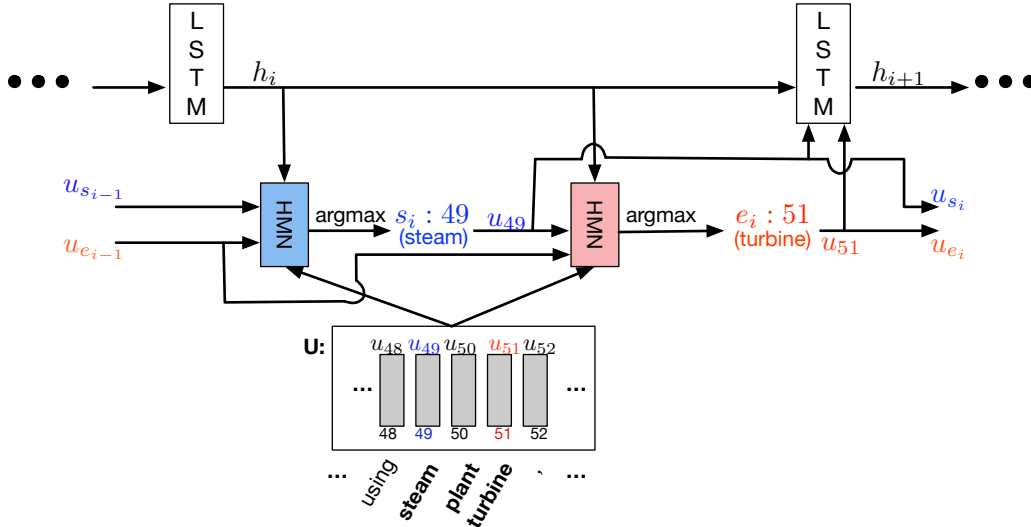

Figure 3: Dynamic Decoder. Blue denotes the variables and functions related to estimating the start position whereas red denotes the variables and functions related to estimating the end position.

Given the current hidden state $h_i$, previous start position $u_{s_{i-1}}$, and previous end position $u_{e_{i-1}}$, we estimate the current start position and end position via Eq. 6 and Eq. 7.

$$s_i = \operatorname*{argmax}_t (\alpha_1, \ldots, \alpha_m) \tag{6}$$

$$e_i = \operatorname*{argmax}_t (\beta_1, \ldots, \beta_m) \tag{7}$$

where $\alpha_t$ and $\beta_t$ represent the start score and end score corresponding to the $t$th word in the document. We compute $\alpha_t$ and $\beta_t$ with separate neural networks. These networks have the same architecture but do not share parameters.

Based on the strong empirical performance of Maxout Networks (Goodfellow et al., 2013) and Highway Networks (Srivastava et al., 2015), especially with regards to deep architectures, we propose a Highway Maxout Network (HMN) to compute $\alpha_t$ as described by Eq. 8. The intuition behind using such model is that the QA task consists of multiple question types and document topics. These variations may require different models to estimate the answer span. Maxout provides a simple and effective way to pool across multiple model variations.

$$\alpha_t = \mathrm{HMN}_{start}\left(u_t, h_i, u_{s_{i-1}}, u_{e_{i-1}}\right) \tag{8}$$

Here, $u_t$ is the coattention encoding corresponding to the $t$th word in the document. $\mathrm{HMN}_{start}$ is illustrated in Figure 4. The end score, $\beta_t$, is computed similarly to the start score $\alpha_t$, but using a separate $\mathrm{HMN}_{end}$.

We now describe the HMN model:

$$
\begin{aligned}
\mathrm{HMN}\left(u_t, h_i, u_{s_{i-1}}, u_{e_{i-1}}\right) &= \max\left(W^{(3)}\left[m_t^{(1)}; m_t^{(2)}\right] + b^{(3)}\right) & (9)\\
r &= \tanh\left(W^{(D)}\left[h_i; u_{s_{i-1}}; u_{e_{i-1}}\right]\right) & (10)\\
m_t^{(1)} &= \max\left(W^{(1)}\left[u_t; r\right] + b^{(1)}\right) & (11)\\
m_t^{(2)} &= \max\left(W^{(2)} m_t^{(1)} + b^{(2)}\right) & (12)
\end{aligned}
$$

where $r \in \mathbb{R}^{\ell}$ is a non-linear projection of the current state with parameters $W^{(D)} \in \mathbb{R}^{\ell \times 5\ell}$, $m_t^{(1)}$ is the output of the first maxout layer with parameters $W^{(1)} \in \mathbb{R}^{p \times \ell \times 3\ell}$ and $b^{(1)} \in \mathbb{R}^{p \times \ell}$, and $m_t^{(2)}$ is the output of the second maxout layer with parameters $W^{(2)} \in \mathbb{R}^{p \times \ell \times \ell}$ and $b^{(2)} \in \mathbb{R}^{p \times \ell}$. $m_t^{(1)}$ and $m_t^{(2)}$ are fed into the final maxout layer, which has parameters $W^{(3)} \in \mathbb{R}^{p \times 1 \times 2\ell}$, and $b^{(3)} \in \mathbb{R}^p$. $p$ is the pooling size of each maxout layer. The max operation computes the maximum value over the first dimension of a tensor. We note that there is highway connection between the output of the first maxout layer and the last maxout layer.

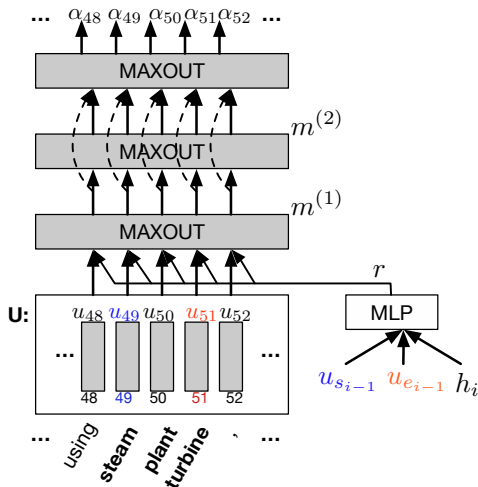

Figure 4: Highway Maxout Network. Dotted lines denote highway connections.

To train the network, we minimize the cumulative softmax cross entropy of the start and end points across all iterations. The iterative procedure halts when both the estimate of the start position and the estimate of the end position no longer change, or when a maximum number of iterations is reached. Details can be found in Section 4.1

## 3 RELATED WORK

**Statistical QA** Traditional approaches to question answering typically involve rule-based algorithms or linear classifiers over hand-engineered feature sets. Richardson et al. (2013) proposed two baselines, one that uses simple lexical features such as a sliding window to match bags of words, and another that uses word-distances between words in the question and in the document. Berant et al. (2014) proposed an alternative approach in which one first learns a structured representation of the entities and relations in the document in the form of a knowledge base, then converts the question to a structured query with which to match the content of the knowledge base. Wang et al. (2015) described a statistical model using frame semantic features as well as syntactic features such as part of speech tags and dependency parses. Chen et al. (2016) proposed a competitive statistical baseline using a variety of carefully crafted lexical, syntactic, and word order features.

**Neural QA** Neural attention models have been widely applied for machine comprehension or question-answering in NLP. Hermann et al. (2015) proposed an AttentiveReader model with the release of the CNN/Daily Mail cloze-style question answering dataset. Hill et al. (2016) released another dataset steming from the children's book and proposed a window-based memory network. Kadlec et al. (2016) presented a pointer-style attention mechanism but performs only one attention step. Sordoni et al. (2016) introduced an iterative neural attention model and applied it to cloze-style machine comprehension tasks.

Recently, Rajpurkar et al. (2016) released the SQuAD dataset. Different from cloze-style queries, answers include non-entities and longer phrases, and questions are more realistic. For SQuAD, Wang & Jiang (2016b) proposed an end-to-end neural network model that consists of a Match-LSTM encoder, originally introduced in Wang & Jiang (2016a), and a pointer network decoder (Vinyals et al., 2015); Yu et al. (2016) introduced a dynamic chunk reader, a neural reading comprehension model that extracts a set of answer candidates of variable lengths from the document and ranks them to answer the question.

Lu et al. (2016) proposed a hierarchical co-attention model for visual question answering, which achieved state of the art result on the COCO-VQA dataset (Antol et al., 2015). In (Lu et al., 2016), the co-attention mechanism computes a conditional representation of the image given the question, as well as a conditional representation of the question given the image.

Inspired by the above works, we propose a dynamic coattention model (DCN) that consists of a novel coattentive encoder and dynamic decoder. In our model, instead of estimating the start and end positions of the answer span in a single pass (Wang & Jiang, 2016b), we iteratively update the

| Model | Dev EM | Dev F1 | Test EM | Test F1 |
|---|---|---|---|---|
| *Ensemble* | | | | |
| DCN (Ours) | **70.3** | **79.4** | **71.2** | **80.4** |
| Microsoft Research Asia * | – | – | 69.4 | 78.3 |
| Allen Institute * | 69.2 | 77.8 | 69.9 | 78.1 |
| Singapore Management University * | 67.6 | 76.8 | 67.9 | 77.0 |
| Google NYC * | 68.2 | 76.7 | – | – |
| *Single model* | | | | |
| DCN (Ours) | 65.4 | **75.6** | **66.2** | **75.9** |
| Microsoft Research Asia * | 65.9 | 75.2 | 65.5 | 75.0 |
| Google NYC * | **66.4** | 74.9 | – | – |
| Singapore Management University * | – | – | 64.7 | 73.7 |
| Carnegie Mellon University * | – | – | 62.5 | 73.3 |
| Dynamic Chunk Reader (Yu et al., 2016) | 62.5 | 71.2 | 62.5 | 71.0 |
| Match-LSTM (Wang & Jiang, 2016b) | 59.1 | 70.0 | 59.5 | 70.3 |
| Baseline (Rajpurkar et al., 2016) | 40.0 | 51.0 | 40.4 | 51.0 |
| Human (Rajpurkar et al., 2016) | 81.4 | 91.0 | 82.3 | 91.2 |

Table 1: Leaderboard performance at the time of writing (Nov 4 2016). * indicates that the model used for submission is unpublished. − indicates that the development scores were not publicly available at the time of writing.

start and end positions in a similar fashion to the Iterative Conditional Modes algorithm (Besag, 1986).

## 4 EXPERIMENTS

### 4.1 IMPLEMENTATION DETAILS

We train and evaluate our model on the SQuAD dataset. To preprocess the corpus, we use the tokenizer from Stanford CoreNLP (Manning et al., 2014). We use as GloVe word vectors pre-trained on the 840B Common Crawl corpus (Pennington et al., 2014). We limit the vocabulary to words that are present in the Common Crawl corpus and set embeddings for out-of-vocabulary words to zero. Empirically, we found that training the embeddings consistently led to overfitting and subpar performance, and hence only report results with fixed word embeddings.

We use a max sequence length of 600 during training and a hidden state size of 200 for all recurrent units, maxout layers, and linear layers. All LSTMs have randomly initialized parameters and an initial state of zero. Sentinel vectors are randomly initialized and optimized during training. For the dynamic decoder, we set the maximum number of iterations to 4 and use a maxout pool size of 16. We use dropout to regularize our network during training (Srivastava et al., 2014), and optimize the model using ADAM (Kingma & Ba, 2014). All models are implemented and trained with Chainer (Tokui et al., 2015).

### 4.2 RESULTS

Evaluation on the SQuAD dataset consists of two metrics. The exact match score (EM) calculates the exact string match between the predicted answer and a ground truth answer. The F1 score calculates the overlap between words in the predicted answer and a ground truth answer. Because a document-question pair may have several ground truth answers, the EM and F1 for a document-question pair is taken to be the maximum value across all ground truth answers. The overall metric is then computed by averaging over all document-question pairs. The offical SQuAD evaluation is hosted on CodaLab [2]. The training and development sets are publicly available while the test set is withheld.

---

[2]https://worksheets.codalab.org

The performance of the Dynamic Coattention Network on the SQuAD dataset, compared to other submitted models on the leaderboard [3], is shown in Table 1. At the time of writing, our single-model DCN ranks first at 66.2% exact match and 75.9% F1 on the test data among single-model submissions. Our ensemble DCN ranks first overall at 71.6% exact match and 80.4% F1 on the test data.

The DCN has the capability to estimate the start and end points of the answer span multiple times, each time conditioned on its previous estimates. By doing so, the model is able to explore local maxima corresponding to multiple plausible answers, as is shown in Figure 5.

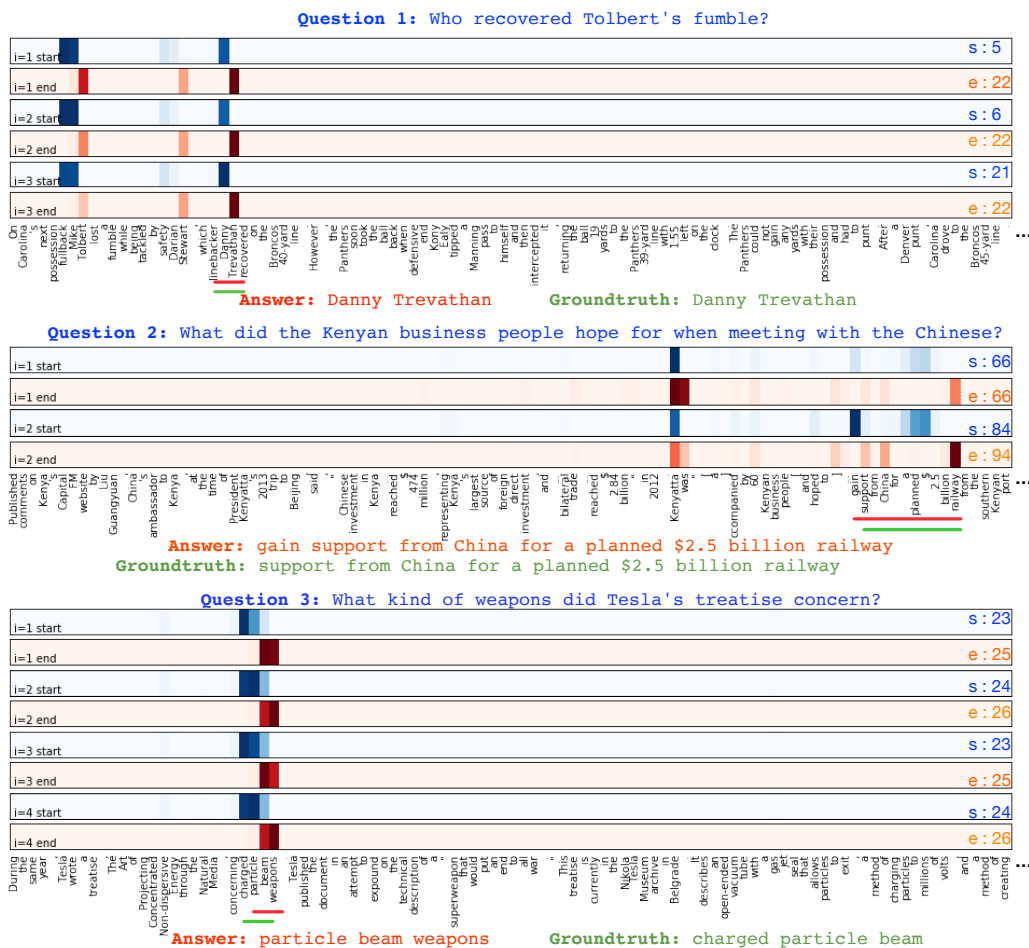

Figure 5: Examples of the start and end conditional distributions produced by the dynamic decoder. Odd (blue) rows denote the start distributions and even (red) rows denote the end distributions. $i$ indicates the iteration number of the dynamic decoder. Higher probability mass is indicated by darker regions. The offset corresponding to the word with the highest probability mass is shown on the right hand side. The predicted span is underlined in red, and a ground truth answer span is underlined in green.

For example, Question 1 in Figure 5 demonstrates an instance where the model initially guesses an incorrect start point and a correct end point. In subsequent iterations, the model adjusts the start point, ultimately arriving at the correct start point in iteration 3. Similarly, the model gradually shifts probability mass for the end point to the correct word.

Question 2 shows an example in which both the start and end estimates are initially incorrect. The model then settles on the correct answer in the next iteration.

---

[3]https://rajpurkar.github.io/SQuAD-explorer

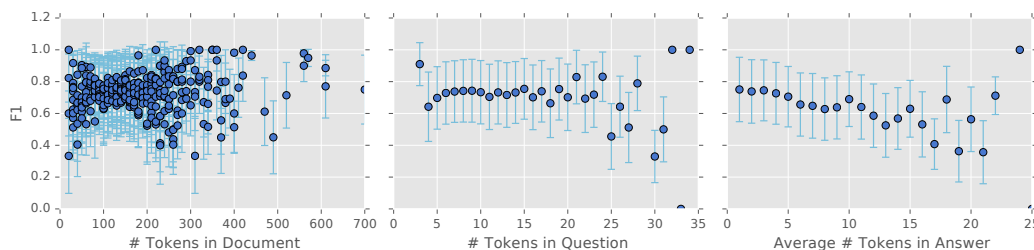

Figure 6: Performance of the DCN for various lengths of documents, questions, and answers. The blue dot indicates the mean F1 at given length. The vertical bar represents the standard deviation of F1s at a given length.

While the dynamic nature of the decoder allows the model to escape initial local maxima corresponding to incorrect answers, Question 3 demonstrates a case where the model is unable to decide between multiple local maxima despite several iterations. Namely, the model alternates between the answers "charged particle beam" and "particle beam weapons" indefinitely. Empirically, we observe that the model, trained with a maximum iteration of 4, takes 2.7 iterations to converge to an answer on average.

**Model Ablation** The performance of our model and its ablations on the SQuAD development set is shown in Table 2. On the decoder side, we experiment with various pool sizes for the HMN maxout layers, using a 2-layer MLP instead of a HMN, and forcing the HMN decoder to a single iteration. Empirically, we achieve the best performance on the development set with an iterative HMN

| Model | Dev EM | Dev F1 |
|---|---|---|
| *Dynamic Coattention Network (DCN)* | | |
| pool size 16 HMN | **65.4** | **75.6** |
| pool size 8 HMN | 64.4 | 74.9 |
| pool size 4 HMN | 65.2 | 75.2 |
| DCN with 2-layer MLP instead of HMN | 63.8 | 74.4 |
| DCN with single iteration decoder | 63.7 | 74.0 |
| DCN with Wang & Jiang (2016b) attention | 63.7 | 73.7 |

Table 2: Single model ablations on the development set.

with pool size 16, and find that the model consistently benefits from a deeper, iterative decoder network. The performance improves as the number of maximum allowed iterations increases, with little improvement after 4 iterations. On the encoder side, replacing the coattention mechanism with an attention mechanism similar to Wang & Jiang (2016b) by setting $C^D$ to $QA^D$ in equation 3 results in a 1.9 point F1 drop. This suggests that, at an additional cost of a softmax computation and a dot product, the coattention mechanism provides a simple and effective means to better encode the document and question sequences. Further studies, such as performance without attention and performance on questions requiring different types of reasoning can be found in the appendix.

**Performance across length** One point of interest is how the performance of the DCN varies with respect to the length of document. Intuitively, we expect the model performance to deteriorate with longer examples, as is the case with neural machine translation (Luong et al., 2015). However, as in shown in Figure 6, there is no notable performance degradation for longer documents and questions contrary to our expectations. This suggests that the coattentive encoder is largely agnostic to long documents, and is able to focus on small sections of relevant text while ignoring the rest of the (potentially very long) document. We do note a performance degradation with longer answers. However, this is intuitive given the nature of the evaluation metric. Namely, it becomes increas-

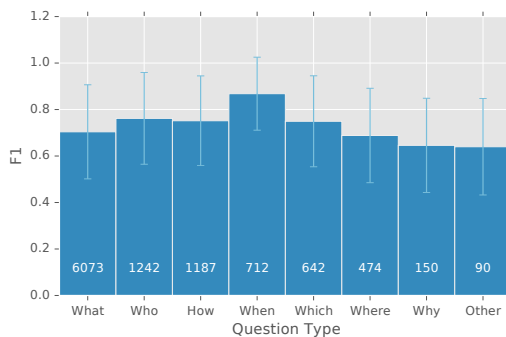

Figure 7: Performance of the DCN across question types. The height of each bar represents the mean F1 for the given question type. The lower number denotes how many instances in the dev set are of the corresponding question type.

ingly challenging to compute the correct word
span as the number of words increases.

**Performance across question type** Another natural way to analyze the performance of the model
is to examine its performance across question types. In Figure 7, we note that the mean F1 of DCN
exceeds those of previous systems (Wang & Jiang, 2016b; Yu et al., 2016) across all question types.
The DCN, like other models, is adept at "when" questions and struggles with the more complex
"why" questions.

**Breakdown of F1 distribution** Finally, we note that the DCN performance is highly bimodal. On
the development set, the model perfectly predicts (100% F1) an answer for 62.2% of examples and
predicts a completely wrong answer (0% F1) for 16.3% of examples. That is, the model picks out
partial answers only 21.5% of the time. Upon qualitative inspections of the 0% F1 answers, some of
which are shown in Appendix A.4, we observe that when the model is wrong, its mistakes tend to
have the correct "answer type" (eg. person for a "who" question, method for a "how" question) and
the answer boundaries encapsulate a well-defined phrase.

## 5 CONCLUSION

We proposed the Dynamic Coattention Network, an end-to-end neural network architecture for ques-
tion answering. The DCN consists of a coattention encoder which learns co-dependent representa-
tions of the question and of the document, and a dynamic decoder which iteratively estimates the
answer span. We showed that the iterative nature of the model allows it to recover from initial lo-
cal maxima corresponding to incorrect predictions. On the SQuAD dataset, the DCN achieves the
state of the art results at 75.9% F1 with a single model and 80.4% F1 with an ensemble. The DCN
significantly outperforms all other models.

ACKNOWLEDGMENTS

We thank Kazuma Hashimoto and Bryan McCann for their help and insights.

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

# A APPENDIX

## A.1 PERFORMANCE WITHOUT ATTENTION

In our experiments, we also investigate a model without any attention mechanism. In this model, the encoder is a simple LSTM network that first ingests the question and then ingests the document. The hidden states corresponding to words in the document is then passed to the decoder. This model achieves 33.3% exact match and 41.9% F1, significantly worse than models with attention.

## A.2 SAMPLES REQUIRING DIFFERENT TYPES OF REASONING

We generate predictions for examples requiring different types of reasoning, given by Rajpurkar et al. (2016). Because this set of examples is very limited, they do not conclusively demonstrate the effectiveness of the model on different types of reasoning tasks. Nevertheless, these examples show that the DCN is a promising architecture for challenging question answering tasks including those that involve reasoning over multiple sentences.

WHAT IS THE RANKINE CYCLE SOMETIMES CALLED?

The Rankine cycle is sometimes referred to as a practical Carnot cycle because, when an efficient turbine is used, the TS diagram begins to resemble the Carnot cycle.

**Type of reasoning** Lexical variation (synonymy)

**Ground truth** practical Carnot cycle

**Prediction** practical Carnot cycle

WHICH TWO GOVERNING BODIES HAVE LEGISLATIVE VETO POWER?

While the Commision has a monopoly on initiating legislation, the European Parliament and the Council of the European Union have powers of amendment and veto during the legislative progress.

**Type of reasoning** Lexical variation (world knowledge)

**Ground truth** the European Parliament and the Council of the European Union

**Prediction** European Parliament and the Council of the European Union

WHAT SHAKESPEARE SCHOLAR IS CURRENTLY ON THE UNIVERSITYS FACULTY?

Current faculty include the anthropologist Marshall Sahlins, historian Dipesh Chakrabarty, ... Shakespeare scholar David Bevington, and renowned political scientists John Mearsheimer and Robert Pape.

**Type of reasoning** Syntactic variation

**Ground truth** David Bevington

**Prediction** David Bevington

WHAT COLLECTION DOES THE V&A THEATRE & PERFORMANCE GALLERIES HOLD?

The V&A Theatre & Performance galleries, formerly the Theatre Museum, opened in March 2009. The collections are stored by the V&A, and are available for research, exhibitions and other shows. They hold the UK's biggest national collection of material about live performance in the UK since Shakespeare's day, covering drama, dance, musical theatre, circus, music hall, rock and pop, and most other forms of live entertainment.

**Type of reasoning** Multiple sentence reasoning

**Ground truth** Material about live performance

**Prediction** UK's biggest national collection of material about live performance in the UK since Shakespeare's day

WHAT IS THE MAIN GOAL OF CRIMINAL PUNISHMENT OF CIVIL DISOBEDIENTS?

**Type of reasoning** Ambiguous

Along with giving the offender his "just deserts", achieving crime control via incapacitation and deterrence is a major goal of crime punishment.

**Ground truth** achieving crime control via incapacitation and deterrence

**Prediction** achieving crime control via incapacitation and deterrence

A.3    SAMPLES OF **CORRECT** SQUAD PREDICTIONS BY THE DYNAMIC COATTENTION NETWORK

HOW DID THE MONGOLS ACQUIRE CHINESE PRINTING TECHNOLOGY?

**ID**: 572882242ca10214002da420

The Mongol rulers patronized the Yuan printing industry. Chinese printing technology was transferred to the Mongols through Kingdom of Qocho and Tibetan intermediaries. Some Yuan documents such as Wang Zhen's Nong Shu were printed with earthenware movable type, a technology invented in the 12th century. However, most published works were still produced through traditional block printing techniques. The publication of a Taoist text inscribed with the name of Tregene Khatun, gedei's wife, is one of the first printed works sponsored by the Mongols. In 1273, the Mongols created the Imperial Library Directorate, a government-sponsored printing office. The Yuan government established centers for printing throughout China. Local schools and government agencies were funded to support the publishing of books.

**Ground truth** through Kingdom of Qocho and Tibetan intermediaries

**Prediction:** through Kingdom of Qocho and Tibetan intermediaries

WHO APPOINTS ELDERS?

**ID** 5730d473b7151e1900c0155b

Elders are called by God, affirmed by the church, and ordained by a bishop to a ministry of Word, Sacrament, Order and Service within the church. They may be appointed to the local church, or to other valid extension ministries of the church. Elders are given the authority to preach the Word of God, administer the sacraments of the church, to provide care and counseling, and to order the life of the church for ministry and mission. Elders may also be assigned as District Superintendents, and they are eligible for election to the episcopacy. Elders serve a term of 23 years as provisional Elders prior to their ordination.

**Ground truth** bishop, the local church

**Prediction** a bishop

AN ALGORITHM FOR X WHICH REDUCES TO C WOULD ALLOW US TO DO WHAT?

**ID** 56e1ce08e3433e14004231a6

This motivates the concept of a problem being hard for a complexity class. A problem X is hard for a class of problems C if every problem in C can be reduced to X. Thus no problem in C is harder than X, since an algorithm for X allows us to solve any problem in C. Of course, the notion of hard problems depends on the type of reduction being used. For complexity classes larger than P, polynomial-time reductions are commonly used. In particular, the set of problems that are hard for NP is the set of NP-hard problems.

**Ground truth** solve any problem in C

**Prediction** solve any problem in C

HOW MANY GENERAL QUESTIONS ARE AVAILABLE TO OPPOSITION LEADERS?

**ID** 572fd7b8947a6a140053cd3e

Parliamentary time is also set aside for question periods in the debating chamber. A "General Question Time" takes place on a Thursday between 11:40 a.m. and 12 p.m. where members can direct questions to any member of the Scottish Government. At 2.30pm, a 40-minute long themed "Question Time" takes place, where members can ask questions of ministers in departments that are selected for questioning that sitting day, such as health and justice or education and transport. Between 12 p.m. and 12:30 p.m. on Thursdays, when Parliament is sitting, First Minister's Question Time takes place. This gives members an opportunity to question the First Minister directly on issues under their jurisdiction. Opposition leaders ask a general question of the First Minister and then supplementary questions. Such a practice enables a "lead-in" to the questioner, who then uses their supplementary question to ask the First Minister any issue. The four general questions available to opposition leaders are:

**Ground truth** four

**Prediction** four

WHAT ARE SOME OF THE ACCEPTED GENERAL PRINCIPLES OF EUROPEAN UNION LAW?

**ID** 5726a00cf1498d1400e8e551

The principles of European Union law are rules of law which have been developed by the European Court of Justice that constitute unwritten rules which are not expressly provided for in the treaties but which affect how European Union law is interpreted and applies. In formulating these principles, the courts have drawn on a variety of sources, including: public international law and legal doctrines and principles present in the legal systems of European Union member states and in the jurisprudence of the European Court of Human Rights. Accepted general principles of European Union Law include fundamental rights (see human rights), proportionality, legal certainty, equality before the law and subsidiarity.

**Ground truth** fundamental rights (see human rights), proportionality, legal certainty, equality before the law and subsidiarity

**Prediction** fundamental rights (see human rights), proportionality, legal certainty, equality before the law and subsidiarity

WHY WAS TESLA RETURNED TO GOSPIC?

**ID** 56dfaa047aa994140058dfbd

On 24 March 1879, Tesla was returned to Gospi under police guard for not having a residence permit. On 17 April 1879, Milutin Tesla died at the age of 60 after contracting an unspecified illness (although some sources say that he died of a stroke). During that year, Tesla taught a large class of students in his old school, Higher Real Gymnasium, in Gospi.

**Ground truth** not having a residence permit

**Prediction** not having a residence permit

A.4    SAMPLES OF **INCORRECT** SQUAD PREDICTIONS BY THE DYNAMIC COATTENTION NETWORK

WHAT IS ONE SUPPLEMENTARY SOURCE OF EUROPEAN UNION LAW?

**ID** 5725c3a9ec44d21400f3d506

European Union law is applied by the courts of member states and the Court of Justice of the European Union. Where the laws of member states provide for lesser rights European Union law can be

enforced by the courts of member states. In case of European Union law which should have been transposed into the laws of member states, such as Directives, the European Commission can take proceedings against the member state under the Treaty on the Functioning of the European Union. The European Court of Justice is the highest court able to interpret European Union law. Supplementary sources of European Union law include case law by the Court of Justice, international law and general principles of European Union law.

**Ground truth** international law

**Prediction** case law by the Court of Justice

**Comment** The prediction produced by the model is correct, however it was not selected by Mechanical Turk annotators.

WHO DESIGNED THE ILLUMINATION SYSTEMS THAT TESLA ELECTRIC LIGHT & MANUFACTURING INSTALLED?

**ID** 56e0d6cf231d4119001ac424

After leaving Edison's company Tesla partnered with two businessmen in 1886, Robert Lane and Benjamin Vail, who agreed to finance an electric lighting company in Tesla's name, Tesla Electric Light & Manufacturing. The company installed electrical arc light based illumination systems designed by Tesla and also had designs for dynamo electric machine commutators, the first patents issued to Tesla in the US.

**Ground truth** Tesla

**Prediction** Robert Lane and Benjamin Vail

**Comment** The model produces an incorrect prediction that corresponds to people that funded Tesla, instead of Tesla who actually designed the illumination system. Empirically, we find that most mistakes made by the model have the correct type (eg. named entity type) despite not including types as prior knowledge to the model. In this case, the incorrect response has the correct type of person.

CYDIPPID ARE TYPICALLY WHAT SHAPE?

**ID** 57265746dd62a815002e821a

Cydippid ctenophores have bodies that are more or less rounded, sometimes nearly spherical and other times more cylindrical or egg-shaped; the common coastal "sea gooseberry," Pleurobrachia, sometimes has an egg-shaped body with the mouth at the narrow end, although some individuals are more uniformly round. From opposite sides of the body extends a pair of long, slender tentacles, each housed in a sheath into which it can be withdrawn. Some species of cydippids have bodies that are flattened to various extents, so that they are wider in the plane of the tentacles.

**Ground truth** more or less rounded, egg-shaped

**Prediction** spherical

**Comment** Although the mistake is subtle, the prediction is incorrect. The statement "are more or less rounded, sometimes nearly spherical" suggests that the entity is more often "rounded" than "spherical" or "cylindrical" or "egg-shaped" (an answer given by an annotator). This suggests that the model has trouble discerning among multiple intuitive answers due to a lack of understanding of the relative severity of "more or less" versus "sometimes" and "other times".

