# Peer review of "Dynamic Coattention Networks For Question Answering"

_ICLR 2017 — accepted_

[Public Comment · Dirk Weissenborn · 10 Nov 2016]
**Ablation study needed**

The paper introduces several interesting mechanisms for answer extraction in a QA setup. However, I find it really difficult to see where the actual benefits come from compared to related architectures like the work of Wang et al. 2016, which was also submitted to ICLR. I believe much of the improvements stem from the dynamic pointer decoder which itself contains various sub-architectures. For the sake of comparability, I would be interested to see how performances would change when:

- not using co-attention but only attention on the question for each context token, similar to Wang et al.
- exchanging the rather large Highway Maxout Network by something simpler, e.g. omitting m1 and m2 completely to get directly to the output, lowering pool-sizes because they seem rather large as well or using a simple MLP instead of the HMN. The intuition behind using Maxout sounds appealing but it has not been verified in the paper.
- allowing only one iteration for the pointer decoder (i.e., direct prediction)

Thanks for considering these suggestions.

[Public Comment · Sahil Sharma · 27 Nov 2016]
**A few questions about vocabulary size**

Hi,

A sentence from Section 4 reads: "We limit the vocabulary to words that are present in the Common Crawl corpus and set embeddings for out-of-vocabulary words to zero." Suppose GloVe vectors' vocabulary size is k (k ~ 2.2 million). Does this mean that the vocabulary size used in this paper is exactly 1 more than GloVe's vocabulary size (k+1)? Does this mean that effectively every out of vocabulary word is replaced with an

[Official Review · AnonReviewer3 · rating 8 · confidence 4 · 16 Dec 2016]
**Official Review**
clarity 5

This paper proposed a dynamic coattention network for the question answering task with long contextual documents. 
The model is able to encode co-dependent representations of the question and the document, and a dynamic decoder iteratively pointing the potential answer spans to locate the final answer. 

Overall, this is a well-written paper. 
Although the model is a bit complicated (coattention encoder, iterative dynamic pointering decoder and highway maxout network), the intuitions behind and the details of the model are clearly presented. 
Also the performance on the SQuAD dataset is good. 
I would recommend this paper to be accepted.

[Official Review · AnonReviewer1 · rating 8 · confidence 3 · 17 Dec 2016 (modified: 20 Jan 2017)]
**Interesting model but need some more analyses**
soundness 5 · originality 5 · clarity 5 · recommendation (unofficial) 5

Summary: The paper proposes a novel deep neural network architecture for the task of question answering on the SQuAD dataset. The model consists of two main components -- coattention encoder and dynamic pointer decoder. The encoder produces attention over the question as well as over the document in parallel and thus learns co-dependent representations of the question and the document. The decoder predicts the starting and the end token of the answer iteratively, with the motivation that multiple iterations will help the model escape local maxima and thus will reduce the errors made by the model. The proposed model achieved state-of-art result on SQuAD dataset at the time of writing the paper. The paper reports some analyses of the results such as performance across question types, document, question, answer lengths, etc. The paper also performs some ablation studies such as performing only single round of iteration on decoder, etc.

Strengths:

1. The paper is well-motivated with two main motivations -- co-attending to the document and the question, and iteratively producing the answer.

2. The proposed model architecture is novel and the design choices made seem reasonable.

3. The experiments show that the proposed model outperforms the existing model (at the time of writing the paper) on the SQuAD dataset by significant margin.

4. The analyses of the results and the ablation studies performed (as per someone's request) provide insights into the various modelling design choices made.

Weaknesses/Questions/Suggestions:

1. In order to gain insights into how much each additional iteration in the decoder help, I would like to see the following -- for every iteration report the mean F1 for questions that converged in that iteration along with the number of questions that converged in that iteration.

2. Example of Question 3 in figure 5 is an interesting example where the model is unable to decide between multiple local maxima despite several iterations. Could authors please report how often this happens?

3. In order to estimate how much modelling of attention in the encoder helps, it would be good if authors could report the performance of the model when attention is not modeled at all in the encoder (neither over question, nor over document).

4. I would like to see the variation in the performance of the proposed model for questions that require different types of reasoning (table 3 in SQuAD paper). This would provide insights into what are the strengths and weaknesses of the proposed model w.r.t the type reasoning required.

5. In Wang and Jiang (2016), the attention is predicted over question for each word in the document. But in table 2, when performing ablation study to make the proposed model similar to Wang and Jiang, C^D is set to C^Q. But isn’t C^Q attention over document for each word in the question? So, how is this similar to Wang and Jiang’s attention? I think QA^D will be similar to Wang and Jiang's attention since QA^D is attention over question for each word in the document. Please clarify.

6. In section 2.1, “n” and “m” are swapped when explaining the Document and Question encoding matrix. Please fix it.

Review Summary: The paper presents a novel and interesting model for the task of question answering on SQuAD dataset and shows that the model outperforms existing models. However, to gain more insights into the functioning of the model, I would like see more analyses of the results and one more ablation study (see weaknesses section above).

[Official Review · AnonReviewer2 · rating 8 · confidence 4 · 23 Dec 2016]
clarity 5 · impact 4

Paper Summary: 
The paper introduces a question answering model called Dynamic Coattention Network (DCN). It extracts co-dependent representations of the document and question, and then uses an iterative dynamic pointing decoder to predict an answer span. The proposed model achieves state-of-the-art performance, outperforming all published models.

Paper Strengths: 
-- The proposed model introduces two new concepts to QA models -- 1) using attention in both directions, and 2) a dynamic decoder which iterates over multiple answer spans until convergence or maximum number of iterations.
-- The paper also presents ablation study of the proposed model which shows the importance of their design choices.
-- It is interesting to see the same idea of co-attention performing well in 2 different domains -- Visual Question Answering and machine reading comprehension.
-- The performance breakdown over document and question lengths (Figure 6) strengthens the importance of attention for QA task.
-- The proposed model achieves state-of-the-art result on SQuAD dataset.
-- The model architecture has been clearly described.

Paper Weaknesses / Future Thoughts: 
-- The paper provides model's performance when the maximum number of iterations is 1 and 4. I would like to see how the performance of the model changes with the number of iterations, i.e., the model performance when that number is 2 and 3. Is there a clear trend? What type of questions is the model able to get correct with more iterations?
-- As with many deep learning approaches, the overall architecture seems quite complex, and the design choices seem to be driven by performance numbers. As future work, authors might try to analyze qualitative advantages of different choices in the proposed model. What type of questions are correctly answered because of co-attention mechanism instead of attention in a single direction, when using Maxout Highway Network instead of a simple MLP, etc?

Preliminary Evaluation: 
Novel and state-of-the-art question answering approach. Model is clearly described in detail. In my thoughts, a clear accept.

[Public Comment · Radu Kopetz · 24 Jan 2017]
**Implementation details**

Thanks a lot for a great paper. Could you please provide some implementation details ?
1. What is the sentinel initialized with ? Is is just a 0 filled vector ? 
2. What are the initial start / end values that you feed to LSTM state before the first iteration ?
3. What initialization values you use for W's and bias ?
4. In the ablation study, you evaluated two layers of MLP. Is that applied on each word individually ? Or on the whole sequence ? In other words, the first layer matrix dimension is [2l, 2l] or [2l*m, 2l*m] ?
5. You mention using dropout. Is is used only on the question/document encodding ? And what amount ? Do you also use L2 regularization ?

Thanks.

[Author Response · Victor Zhong · 14 Feb 2017]
**Update**

We sincerely thank all reviewers for your feedback and comments! We have updated our paper, making minor adjustments such as fixing typos. We have also included some additional ablation studies, requested during the review process, in the appendix.

[Final Decision · Program Chairs · 06 Feb 2017]
**ICLR committee final decision**

The program committee appreciates the authors' response to concerns raised in the reviews. All reviewers agree that this is a good piece of work that should be accepted to ICLR. Authors are encouraged to incorporate reviewer feedback to further strengthen the paper.